# Patient Experiences and Perspectives of Their Decision-Making to Accept Lung Transplantation Referral: A Qualitative Study

**DOI:** 10.3390/ijerph20054599

**Published:** 2023-03-05

**Authors:** Mengjie Chen, Xueqiong Zou, Jiang Nan, Baiyila Nuerdawulieti, Xiahere Huxitaer, Yuyu Jiang

**Affiliations:** Research Office of Chronic Disease Management and Rehabilitation, Nursing Department, Wuxi School of Medicine, Jiangnan University, No.1800 Lihu Avenue, Wuxi 214122, China

**Keywords:** decision-making, lung transplantation, qualitative research, referral

## Abstract

Providing early lung transplantation referral services should be considered to reduce pre-transplant mortality in patients with advanced disease. This study aimed to explore the reasons for lung transplantation referral decisions in patients and provide evidence for the development of transplantation referral services. This was a qualitative, retrospective, and descriptive study involving conventional content analysis. Patients in evaluation, listing, and post-transplant stages were interviewed. A total of 35 participants (25 male and 10 female) were interviewed. Four main themes were defined: (1) expectations for lung transplantation leading to the decision (a gamble for a silver lining, a return to normal life, and occupation); (2) facing uncertain outcomes (personal luck arranging everything; belief in success; incidents leading to “It’s decided then”; hesitation when making a decision due to fear); (3) facing various information from peers, doctors, and so on; (4) complex policy and societal support (providing transplantation referral services earlier, family attachment and oral approval responses contribute to the referral decision, and so on). The findings of this study might enrich existing referral services, including training for family members and healthcare providers, a checklist and package of critical events in the patient lung transplantation referral decision-making process, precision services based on behavioral personas, and a curriculum to enhance patient decision self-efficacy.

## 1. Introduction

According to the latest statistics from the International Society for Heart and Lung Transplantation (ISHLT), 69,200 adult patients registered with ISHLT underwent lung transplantation from 1992 to 2018 [1]. The data did not include all lung transplantation centers worldwide, and the actual figure was higher [2]. Lung transplantation has been the most reported organ transplantation in the last decade [3]. The procedure of lung transplantation is complex, including referral, evaluation, and listing. According to the published data [4], the waiting time for United States candidates has followed a downward trend. The median time was 42 days in 2020 [4]. The pre-transplant mortality rate increased by 16.1% [4]. Related studies in both the United States [5] and France [6] have shown that improved transplantation referral strategies can prevent some patients from dying before transplantation.

Lung transplantation referral is the beginning of the entire lung transplantation process. The ISHLT guidelines have suggested that lung transplantation referral is a complex process and a worldwide challenge [7]. Listing is the stage after referral. Late referrals may lead to the death of patients in the listing stage [5]. A late referral is thought to be associated with pre-transplant mortality [5,6]. The factors contributing to late referrals include the following: Insurance status, drive time to the nearest lung transplantation center, advanced age, lack of high school education, the timing of lung transplant-related information provision, psychosocial barriers, a patient’s incorrect perception of their disease status, and so forth [5,8,9,10]. Reviews by Stubber et al. [10] and Skogeland et al. [11] pointed out that psychosocial disorders such as guilt and anxiety in patients, as well as the need for intrinsic resources (optimism, hope, etc.) and extrinsic resources (information support, peer support, etc.), are common in the referral and listing stages. It has been suggested that early referral services can not only improve the factors of late referral but also meet the psychological and informational needs of patients in the waiting stage [7,9,10,11]. Early transplantation referral services can give patients a greater opportunity to learn about the risks and benefits of lung transplantation [12]. This increases the possibility of patients being placed on the waiting list [9]. Ramos et al. [5] and Martin et al. [6] found that if well-established early referral services are provided, more than 30% of potential lung transplantation patients could avoid the possibility of pre-transplant death. Some studies have shown decision aids as a strategy to help patients receive early lung transplantation referral services [9,10,11,12,13]. The provision of information plays an important role in this process. To date, evidence on the different attitudes of patients and the use of information during the referral stage is limited. Providing disease-related information alone cannot satisfy the patient’s need for varied information [14]. Current lung transplantation information services also neglect being readily available and personalized.

Until December 2020, the China Lung Transplantation Registry reported 2013 lung transplantations [15]. The lung transplantation process in China is consistent with that of other countries; the three stages before the surgery are referral, evaluation, and listing. China uses the China Organ Transplant Response System to ensure fair, efficient, and safe organ allocation. Aryal and Nathan [16] believed China to be a leader in the field of lung transplantation in Asia; they also presumed that the sharing of China’s lung transplantation experience could contribute to the global development of lung transplantation. No patient-centered studies have reported on the referral decision-making process for Chinese lung transplant patients. Additionally, studies on strategies for providing referral decision support services for potential lung transplantation patients are few.

This study was based on retrospective experiences and perspectives of the lung transplantation referral stage for patients with advanced lung disease (ALD) in China to understand the decision-making process of patients when accepting referrals. This study aimed to explore the reasons behind patients receiving lung transplantation referrals and provide evidence to support the improvement of existing referral service components.

## 2. Materials and Methods

### 2.1. Design

A qualitative, retrospective, and descriptive study was conducted to explore the experiences and perspectives of the referral stage, which occurs prior to listing for transplantation, in patients with ALD. Conventional content analysis was performed to analyze individual semi-structured interviews [17]. As the existing theory is limited in the field of lung transplantation referral decision-making, the findings derived from the conventional content analysis are considered more appropriate to provide suggestions and future research directions for existing referral services [17]. The study followed consolidated criteria for reporting qualitative research (COREQ) to ensure quality and transparency [18].

This study was performed in accordance with the Declaration of Helsinki and approved by the Medical Ethics Committee of Jiangnan University (JNU20210618IRB06).

### 2.2. Recruitment and Participants

A purposive sampling method was used to select participants. The participants were recruited at a lung transplantation center and a district rehabilitation hospital from July 2021 to April 2022. The eligibility criteria for participants were as follows: (1) aged ≥ 18 years; (2) patients were in the evaluation stage, the listing stage, or within 6 months of transplantation [19]; (3) the ability to communicate without severe dyspnea; and (4) the ability to communicate in Chinese. We selected patients of different ages, literacy levels, sexes, and diagnoses to ensure a broad selection.

Participants were informed of the study’s purpose, study procedures, risks and benefits, confidentiality, and compensation, and that participation was voluntary. We assured participants that we would hide their identities by changing their names, changing the names of any other people they mentioned, etc., to ensure anonymity. Consent was provided in writing to confirm a willingness to participate.

### 2.3. Data Collection

At the beginning of the interview, general information about the participants was collected through a self-designed questionnaire, including gender, age, education, marital status, diagnosis, stage in the transplantation process, etc.

Based on a review of the literature on qualitative studies of the experiences and perspectives of patients in the referral, evaluation, and listing stage, the research team worked together to construct a semi-structured interview guideline. The interview guideline was piloted in five patients and was not modified. The pre-interview data are included in the data analysis. Open-ended questions were used to understand the participants’ decision-making for a referral. The interview guideline is shown in Table 1. During the interview, when the participant’s statements were unclear, the interviewer would further confirm the statement.

Semi-structured, face-to-face interviews were used for data collection. The interview time and location were confirmed with participants as soon as they were recruited. Thirty-three patients were interviewed in a quiet hospital reception room. However, two participants were interviewed at the bedside due to their requirement for 24 h oxygen. For them, bed curtains were used to create a separate and quiet space and thus to develop a safe and silent environment for participants. All interviews were conducted without disturbing other patients. No third person was present during the interviews. Three patients refused to participate because of physical discomfort. Written informed consent was obtained from the patients to publish this paper. This study had no impact on the participants’ subsequent processes.

The interview was conducted by the first author of this study: a postgraduate student with 3 years of qualitative research study and practice experience. The interviewer made efforts to establish rapport with the participants before the interview. The whole interview was recorded with a recorder after the patients gave permission. During the interview, the researcher recorded the feelings of the participants and the changes in their moods and movements. The average length of the interviews was 40 min. After 35 interviews, no new themes emerged, indicating data saturation. The data collection was stopped.

### 2.4. Data Analysis

Within 24 h of the interviews, the anonymized recordings were repeatedly and carefully listened to and then transcribed in Chinese verbatim by an undergraduate student who was not involved in this study. The first author and participants verified the correctness and completeness of the transcriptions. Then, the transcriptions were anonymized and imported into NVivo 12 Plus. Participants were numbered according to the interview order (Patients: P1, P2…)

Conventional content analysis was used to code and categorize. Two authors repeated readings and immersion in the material. Two authors separately open-coded, created categories, and conceptualized the material after ensuring complete familiarity with the text and understanding of the data [20]. The disagreements were discussed with a third researcher until all three researchers agreed. A preliminary theme, subthemes, and their concepts were developed after the three researchers reached a consensus. Finally, the themes, subthemes, and concepts were reviewed with some of the participants and a research team of eight researchers to reduce biased interpretations and ensure trustworthiness. The research team included both males and females who had rich experience in qualitative research in respiratory care.

The quotes of the results were translated into English by a research team member and a non-research team member who was proficient in English. Another researcher with experience studying in the United States checked and provided feedback on the translated quotes. The disagreements were discussed until all three people agreed.

## 3. Results

Table 2 shows the general information of the 35 participants. The age of the participants ranged from 30 to 76 years. The mean age was 51.4 years, and the standard deviation was 16.3 years. Further, 25 male and 10 female participants were interviewed. The majority of participants were married (97%). More than half of the participants were diagnosed with idiopathic interstitial pneumonitis (71%). The other participants had pulmonary arterial hypertension (6%), chronic obstructive pulmonary disease (11%), and pneumonoconiosis (11%).

The findings describe the retrospective experiences and perspectives of the referral stage in patients with ALD. Table 3 shows the themes and subthemes. Four main themes and twelve subthemes were defined.

### 3.1. Expectations for Lung Transplantation Leading to the Decision

The majority of participants reported that their expectations of a successful lung transplantation treatment would lead to their referral decision.

#### 3.1.1. A Gamble for a Silver Lining

The majority of participants regarded lung transplantation as a gamble. They believed that a successful surgery meant a new life. On the contrary, a failed surgery meant death. Participants also described that lung transplantation was their chance and last straw.

*It was a gamble! It was a life-or-death gamble! However, that was the only way to survive*. (P33)

*There was no other way out under the circumstances…There was still a chance if I accepted the referral*. (P3)

*I was so desperate that stuck at home and waited to die… In desperation, I grasped at the final chance*. (P34)

#### 3.1.2. Return to Normal Life

The majority of participants described that they were eager to achieve a “normal life.” For them, normal life meant traveling as in the past, having a high quality of life, undertaking simple household chores, and holding grandchildren.

*I still traveled when I was not very ill… If the operation is successful, I will continue to travel*. (P23)

*You can’t walk or even breathe… You can’t do anything but stay in hospital. It affects the quality of life too much*. (P20)

*I just want to do some basic housework if I can pull through*. (P9)

*I said I hadn’t seen my son get married yet. And I also want to hold my grandchildren*… (P35)

#### 3.1.3. Return to Occupation

A minority of participants indicated that they wished to return to work so they could realize their own worth and support their families.

*My classmates who graduated at the same time are still working. I want to work like them to realize my social value*. (P30)

*I have two young children to take care of. I need to get back to work to support my family*. (P5)

### 3.2. Facing Uncertain Outcomes

The majority of participants expressed a fear of uncertain outcomes. Some participants did not fear uncertain outcomes and expressed their determination in decision-making. However, some participants expressed a fear of uncertain outcomes and reported hesitation in their decision-making process.

#### 3.2.1. Personal Luck Arranging Everything

A minority of participants believed that the success of lung transplantation was related to personal luck and that they did not need to worry much about uncertain outcomes.

*If you’re lucky, you’ll be better off, but if you’re not, you might not be. It’s destiny… I don’t feel scared anymore*. (P10)

*The success rate of surgery, survival rate, and so on have a lot to do with luck. Fear is useless*. (P13)

#### 3.2.2. Belief in Success

A minority of participants mentioned that they were young and confident about their own physical condition. They believed that their lung transplantation surgery and postoperative recovery would go well.

*I feel like I was in a better physical condition and younger than other people who also need transplantation. I believed that I would get better*. (P7)

A minority of participants who had many successful experiences had a high level of confidence.

*It’s like when I used to work part-time. Everything that I decided to do with determination was always finished successfully… I am also confident in the decision of lung transplantation*. (P2)

#### 3.2.3. Incidents Leading to “It’s Decided Then”

The majority of participants reported that they could make the decision calmly after some incidents. Participants reported two types of incidents.

A minority of participants described that they could accept any outcome calmly after submitting a referral application.

*You had submitted a referral application. It’s decided then. There is not much of a point in worrying too much, you just wait for it*. (P8)

The majority of participants described that “getting lung transplantation suggestion from their doctor” was an incident which enabled them to make the decision calmly.

*I didn’t know whether I should make this decision…On the contrary, when my doctor gave me the lung transplantation suggestion, I calmly knew it’s time to decide*. (P6)

*I need a doctor to tell me it’s time to have this operation and then I can make the decision since it’s a critical thing for me*. (P24)

#### 3.2.4. Hesitation When Making the Decision Due to Fear

A minority of participants described their hesitation due to fear. Participants indicated that they might finally refuse the lung transplantation, even if they decided to accept the referral at first and had been placed on the waiting list.

*After all, no one can be sure of the surgical success rate… I would often cry with fear. I don’t know if I would stick to that option*. (P32)

*I’m afraid I’ll only live five more years or less. I gave up once one year ago because of this when I was on the list*. (P15)

### 3.3. Facing Various Information from Peers, Doctors, and so on

All the participants reported their attitudes toward, and use of, information when they faced various information from peers, doctors, and so on during the decision-making process.

#### 3.3.1. Different Attitudes toward Information

A majority of participants considered information to be extremely important in the decision-making process. They would try to obtain information from the internet, peers, doctors, and so on.

*Everyone needs information when making decisions. I consulted doctors, nurses, patients, as well as learned a lot from the Internet before deciding on a referral*. (P7)

The majority of participants indicated that they preferred to obtain information from their peers rather than from other sources.

*Only people who actually have the same disease can understand each other, so I prefer to get information from peers when I make this decision*. (P4)

*I prefer to communicate with my peers… I learned a lot from my peers in WeChat when discussing it*. (P20)

The majority of participants reported that their peers’ positive experiences of lung transplantation were especially helpful in their decision-making process. Positive experiences could enhance their confidence. However, a minority of participants did not care for peer experiences. They believed that everyone had different conditions.

*I’ve seen so many patients who have recovered, so I think it’s a good idea to take this chance (lung transplantation) and keep going*. (P5)

*Knowing that there were so many patients who had recovered well after the surgery boosted my confidence*. (P6)

*Everyone is different after surgery, so when making a decision, don’t base your decision entirely on how worse others have faced after that*. (P13)

*They said they were worse off after the operation…Can everyone have the same situation? No one can guarantee that the outcome of the surgery will be good*. (P14)

Additionally, a minority of participants mentioned that the information about cures for patients with coronavirus disease 2019 promoted the choice of lung transplantation.

*In Wuhan, coronavirus disease 2019 was so serious… It gave me the hope to be cured like the patients with coronavirus disease… I decided to undergo surgery, and I thought I would be fine too*. (P16)

#### 3.3.2. Using Information to Weigh Risks

The majority of participants used the information to weigh the risks when considering the referral. A minority of participants regarded this weighing of risks as a weighing of cost performance.

*The doctor told me the pros and cons of the lung transplantation… I weighed the success rate and the long-term survival rate. Then, I decided on a referral*. (P6)

*I learned that the technology was constantly being updated and the volume of lung transplants was constantly increasing… It inspired me to consider giving it a shot*. (P9)

*The decision was made after weighing the uncertainty, success rate, survival rate… including the financial support. In fact, I’d like to weigh the cost performance of the lung transplantation*. (P12)

### 3.4. Complex Policy and Societal Support

All the participants reported that policy and societal support had a significant impact on their referral decisions.

#### 3.4.1. Provided Earlier Transplantation Referral Services

A minority of participants reported their unmet need for earlier referral advice.

*When I started treatment for pulmonary fibrosis, no one told me that I would eventually need a transplantation… I need earlier referral advice and have more time to make the decision*. (P24)

A minority of participants also pointed out that early referral services needed to be improved.

*I believe that there is definitely a benefit to having surgery earlier. However, in fact, there are many peers who do not know or do not agree with this…The situation should be improved*. (P20)

#### 3.4.2. Family Attachment and Oral Approval Responses Contributed to the Referral Decision

The majority of participants said that financial support from their families greatly helped them in the decision-making process.

*Without their financial support, I definitely wouldn’t have been able to have the chance to make this decision*. (P11)

A minority of participants said that they were persuaded and touched by family attachment.

*My son said, “Mom, what you should think about is that when I come back from school, I can call my mother at home and that’s all that matters…’’ I was so moved that I made the decision of referral*. (P9)

Additionally, the majority of participants reported that the response of family members to their own decision could influence their decision-making process.

*Yes, they said at the beginning that they would let me decide for myself. However, I still needed their oral response to my final decision. Then, I will feel comfortable and act on my decision*. (P29)

*Although the decision was up to me, my wife was initially skeptical about the lung transplantation and I didn’t insist on the decision. Finally, there was no other way; she agreed, and I came here*. (P24)

A minority of participants also tried to obtain that approval response through roundabout expression.

*My son and daughter asked me, Dad, have you decided yet? I said, if you guys want me to try, I will try and struggle against the disease. On the contrary, I won’t try and I won’t blame you guys… In fact, I had decided to accept the referral at the time they asked me… I just want their approval response…Their approval response can make me feel secure*. (P1)

#### 3.4.3. Obtaining Financial Support from Medical Insurance and Multi-Welfare

The majority of participants said that the government health insurance program and apps for online fundraising campaigns solved most of their financial issues, which made them brave enough to make the decision to refer.

*I didn’t decide to come here until I was covered by this government health insurance program...In fact, I wanted to do it a few years ago, but I couldn’t afford the transplantation. I was afraid to accept transplantation at that time*… (P26)

*You know, on that app, hundreds of people donated to me 5 yuan, 150 yuan, 1000 yuan…everyone donated for my medical care…they saved my life and gave me confidence to accept the referral (voice trembles)*. (P2)

## 4. Discussion

This study explored the reasons behind patients’ decisions to undertake lung transplantation referrals. The patient expectations of lung transplantation, access to information, and policy and social support were the reasons for patients’ referral decisions and the need for early referral services, which is consistent with previous findings [8,9,10]. Additionally, there are four new findings in this study: (1) family attachment and oral approval response, (2) incidents leading to “It’s decided then”, (3) different patient attitudes and use of information, and (4) some strategies that can improve decision-making self-efficacy in the lung transplantation referral decision-making process. The above findings could improve the existing lung transplantation referral service component.

The findings of this study showed that some patients’ decision-making was based on family attachment and oral approval response. In the studies by Macdonald et al. [21] and Yelle et al. [22], only the importance of family companionship in patients was mentioned: there was no discussion of the two new forms of family support mentioned above. The aforementioned family-based decision-making phenomenon is more common in Asian countries [23]. This phenomenon is a reflection of the ideology of Confucianism in China. Confucianism makes it easier for people to hold collectivist views [24]. Collectivists prefer to see themselves as part of a group, so they often involve others in decision-making processes to maintain organizational harmony. Most studies have emphasized the importance of informed decision-making for caregivers [25]. The present study has enriched the knowledge of the role of caregivers in decision-making. When providing health education to caregivers, it is recommended to include training on the expression of family attachment and oral approval response. Future experimental studies are needed to provide more evidence for the improvement of the training content provided to caregivers.

The roundabout expression reported in this study was not only an expression of a high-context culture but also a communication technique for patients to achieve their communication goals [26]. The roundabout expression is an indirect communication. When the patient was asked by family members about the lung transplantation decision, the patient did not respond “yes” or “no” directly, although they had decided to accept the lung transplantation. They expressed they followed their family members’ decision. If their family members agreed, they agreed. If not, they would give up and not blame them. Hammami et al. [27] highlighted that the patients from different contextual cultures had different needs for information disclosure. When healthcare providers provide information to patients using communication that is not culturally appropriate to the patient’s context, it can affect the patient’s perception of the illness and the clinical assessment of the patient’s psychosocial impairment, leading to late referrals [23]. In low-context cultures, communication must be clear and detailed to avoid distortion. In high-context cultures, communication is more focused on interpersonal relationships, social context, and so forth. The generation of different-context cultures is related to factors such as economic methods, population density, history, and traditional culture. Healthcare providers can identify patients’ different communication needs based on the aforementioned factors and provide different communication modes. Current referral services do not mention the need for different communication styles based on contextual culture. Therefore, a study on contextual culture might become a future research direction of lung transplantation referral services, such as developing specific verbal tricks for different contexts and gradually forming communication norms through the application of verbal tricks. Moreover, the training of healthcare providers should be enhanced so that healthcare providers can correctly identify the potential goals and needs of the patients during the communication process and provide personalized assistance to ensure the quality of service.

This study found two incidents that facilitated the patients’ referral decision-making process: one was submitting a referral application, and the other was receiving a lung transplantation suggestion from their doctor. Ivarsson et al. [28] used the critical incident technique (CIT) in a qualitative study to identify the information support, social support, and psychological support needs of patients in the waiting stage. In fact, CIT has been widely used to identify patients’ decision-making needs in health services. Runeson et al. [29] used this technique to explore the factors influencing children’s involvement in the decision-making process. Barradell et al. [30] used this technique to understand the pulmonary rehabilitation decision-making needs of patients with chronic obstructive pulmonary disease. Holden et al. [31] integrated critical incident and fictitious scenario techniques to create three distinct patient self-care decision-making modes to enhance current patient decision assistance programs. However, there are no studies that have used CIT to analyze these incidents that facilitate patients’ lung transplantation referral decisions. Adding CIT to healthcare staff training or a “critical incident technician” to the team could assist healthcare staff in identifying and analyzing these incidents in the patient referral decision-making process. However, the effectiveness of such measures needs to be further verified in quantitative studies. The development of a critical event checklist and package for a lung transplantation referral patient may become a hot topic for future research.

Patients have different attitudes and use various information from peers, doctors, and so on in their decision-making process. Many studies obtained similar results but did not explore the causes and proposed more effective coping strategies [28,32]. The present study attempts to explain this phenomenon in the context of decision behavior, which may be explained by the different decision-making styles of patients. Schwartz et al. [33] indicated two different decision-making styles. Satisfying-style people tend to choose satisfactory decisions without making trade-offs, while maximizing-style people tend to compare each option to pursue the best decision [34]. In recent years, the classification of decision-making styles has usually used quantitative statistical methods [35], qualitative studies [36], or mixture studies [37], and has been used as a basis for developing user personas. The advantage of user personas is that hard-to-find user needs can be identified and used to provide more accurate e-health services [37]. With the global prevalence of COVID-19, online referral services are a blessing for vulnerable patients with ALD. This suggests to us that user persona development will be a future research trend.

This study identified several strategies that can improve decision-making self-efficacy during the patient’s referral decision-making process: awakening confidence from the success of the decision-making processes of patients previously, providing transplantation referral decision experiences from peers, providing family attachment and oral approval responses, improving the ability to cope with lung transplantation uncertainty, and evoking the patient’s expectations for future life. An investigation of patients with colorectal cancer [38] showed that higher decision self-efficacy was associated with lower decisional conflict. It can be seen that self-efficacy interventions can be added to early referral services to decrease decisional conflict, decisional regret, and late referral. Smith et al. [9] proposed measures in terms of making patients aware of the benefits of transplantation and misconceptions about the risks associated with transplantation in order to enhance patient self-efficacy. This supports some of the aforementioned strategies. In addition, the strategies proposed above are presented from additional perspectives, such as peers and family members, expanding on existing strategies to enhance patient self-efficacy in referral decision-making services. In the future, the aforementioned strategies can be transferred into a structured curriculum for patient health education or incorporated into referral decision-making services.

### Strengths and Limitations

This study is the first to explore the referral decision-making process for patients with ADL in China. Guided by a rigorous methodology, this study proposes several corresponding measures around four new findings. These measures will help improve the existing lung transplantation referral service component.

This study has certain limitations. As an exploratory study, the measures proposed in this study still need experimental studies to further verify their effectiveness. The present study ensured data saturation, but the results of this qualitative study still did not cover all transplantation patients. This study only covered patients in China, and hence, these findings might not be applicable to patients in other countries. This study interviewed only patients with ALD without their relatives, healthcare providers, policy implementers, and policymakers. Patient decision-making is influenced by various aspects such as family, healthcare providers, and policies. Further research is required to fully explore the experiences of these staff in the patient decision-making process. As a retrospective study, we selected patients before or within 6 months after transplantation to avoid patient memory bias as much as possible. Additionally, patients were reminded during the interview process to ensure that the experiences and perceptions shared by them were during the referral decision period to reduce the effect of time on the results of this study.

## 5. Conclusions

We explored the reasons for lung transplantation referral decisions among Chinese patients with ALD. Well-established early referral services are a worldwide challenge. A study of the lung transplantation referral decision-making process in Chinese patients can provide useful experience to international lung transplantation referral services.

This study showed that providing personalized early referral decision support was a challenge due to the complexity of the referral decision-making process. We propose strategies to enrich the existing referral services. It was recommended that curricula or educational materials be developed to enhance the self-efficacy of patient decision-making. Additionally, it was recommended that the family members be trained in expressing attachment and the sharing of oral approval responses with patients, and that healthcare providers be trained in identifying the potential needs of patients using communication and critical incident techniques. Further, the study highlights the need for targeted verbal tricks in different-context cultures to be developed and to gradually develop communication norms after application. We pointed out the need to develop a checklist and package for critical events in the lung transplantation referral decision-making process of patients. It is prospective to conduct research on the persona of patient referral decision-making behavior. This can optimize online referral decision-making services. The aforementioned strategies can help potential lung transplant patients consider lung transplant referrals early and reduce pre-transplant mortality.

## Figures and Tables

**Table 1 ijerph-20-04599-t001:** Interview guideline.

Questions
1. How did you learn about lung transplantation?
2. How did you feel when being informed that you needed a referral for lung transplantation? How did you make the decision for a referral? Please describe the whole experience and perceptions.
3. In the referral decision-making process, how did your family respond to you?
4. In the referral decision-making process, what information did you learn? What do you think after obtaining the information?
5. In the referral decision-making process, what did you think of the service and policy?
6. Is there anything else you would like to share about your experience in the referral decision-making process?

**Table 2 ijerph-20-04599-t002:** Characteristics of participants (*n* = 35).

Characteristics	P ^a^ (*n* = 35)
Age (years), mean (SD) (range)	51.4 (16.3) (30–76)
Gender, *n* (%)	
Male	25 (71)
Female	10 (29)
Marital Status, *n* (%)	
Single	1 (3)
Married/cohabiting	34 (97)
Highest level of education completed, *n* (%)	
Elementary school or below	4 (11)
Middle school	11 (31)
High school	12 (34)
Higher education or above	8 (23)
Diagnosis, *n* (%)	
Pulmonary arterial hypertension	2 (6)
Chronic obstructive pulmonary disease	4 (11)
Pneumonoconiosis	4 (11)
Idiopathic interstitial pneumonitis	25 (71)
Stage in transplantation trajectory, *n* (%)	
Evaluation	8 (23)
Listing	9 (26)
Post-transplant	18 (51)

^a^ Patient.

**Table 3 ijerph-20-04599-t003:** Themes and subthemes.

Themes	Subthemes
1. Expectations for lung transplantation leading to the decision	(1) A gamble for a silver lining
	(2) Return to normal life
	(3) Return to occupation
2. Facing uncertain outcomes	(1) Personal luck arranging everything
	(2) Belief in success
	(3) Incidents leading to “It’s decided then”
	(4) Hesitation when making the decision due to fear
3. Facing various information from peers, doctors, and so on	(1) Different attitudes toward information
	(2) Using information to weigh risks
4. Complex policy and societal support	(1) Provided earlier transplantation referral services
	(2) Family attachment and oral approval responses contributed to the referral decision
	(3) Obtaining financial support from medical insurance and multi-welfare

## Data Availability

The data presented in this study are available on request from the corresponding author.

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
