# Peer review of "Patient Experiences and Perspectives of Their Decision-Making to Accept Lung Transplantation Referral: A Qualitative Study"

_ijerph, 2023, doi:10.3390/ijerph20054599_

Round 1

Reviewer 1 Report

Thank you for giving me the opportunity to read this interesting manuscript. It provides the patient’s experiences and perspectives on lung transplantation referral decision-making. The paper is well-written. In my opinion, it is not clear enough what new insights this study offers. There is a range of qualitative studies in this area with similar findings. Nevertheless, I would like to offer the authors the opportunity to revise the manuscript from this perspective and to elaborate on these aspects.

I also have the following more minor comments:

·       For better clarity, I recommend numbering the four topics in the abstract.

·       Why is it relevant that the interviewer was female? This information can be deleted or must be explained.

·       Were the transcriptions anonymous?

·       In Table 3, capitalization is not consistent.

·       All ethics information is missing - which ethics committee approved this? In what way did the patients consent? What level of anonymity did this study have? Was there any impact on the patients' subsequent process?

·       Patient quotes should be more clearly highlighted as such.

·       In the discussion, the authors should point more to comparable studies and place it in previous findings. In this context, it should be emphasized to what extent the study has novelty character.

Author Response

Response to Reviewer 1 Comments

Thank you for giving me the opportunity to read this interesting manuscript. It provides the patient’s experiences and perspectives on lung transplantation referral decision-making. The paper is well-written.

Response: Thank you very much for your kind considerations and careful comments. According to your comments, we have modified the manuscript, especially in terms of being clear about new insights. Here below you can find our responses on a point-by-point basis. We marked page and line number in parentheses.

Point 1: In my opinion, it is not clear enough what new insights this study offers. There is a range of qualitative studies in this area with similar findings. Nevertheless, I would like to offer the authors the opportunity to revise the manuscript from this perspective and to elaborate on these aspects.

Response 1: Thank you very much for giving us an opportunity to clarify our new insights. As you suggested in "Point 8", we added highlighting the relevance of the results to the existing literature in the first paragraph of the discussion section and in each of the discussion points. In addition, we also clarified the innovative points of this study in "Strengths and Limitations". The following is the modified content (The highlighted places are the descriptions that were modified and related to the new insights.):

“This study explored the reasons behind patients' decisions to undertake lung transplantation referrals. The patient expectations of lung transplantation, access to information, and policy and social support were the reasons for patients' referral deci-sions and the need for early referral services, which is consistent with previous find-ings [8-10]. Additionally, there are four new findings in this study: (1) family attach-ment and oral approval response, (2) incidents leading to “It's decided then”, (3) dif-ferent patient attitudes and use of information, and (4) some strategies that can im-prove decision-making self-efficacy in lung transplantation referral decision-making process. The above findings could improve the existing lung transplantation referral service component.”(Page: 9; Line: 335)

“The findings of this study showed that some patients' decision making was based on family attachment and oral approval response. In the studies by Macdonald et al. [21] and Yelle et al. [22], only the importance of family companionship in patients was mentioned: there was no discussion of the two new forms of family support mentioned above.” (Page: 9; Line: 344)

“The roundabout expression reported in this study was not only an expression of a high-context culture but also a communication technique for patients to achieve their communication goals [26]. Hammami et al. [27] highlighted that the patients from different contextual cultures had different needs for information disclosure. When healthcare providers provide information to patients using communication that is not culturally appropriate to the patient's context, it can affect the patient's perception of the illness and the clinical assessment of the patient's psychosocial impairment, leading to late referrals [23]. In low-context cultures, communication must be clear and detailed to avoid distortion. In high-context cultures, communication is more focused on interpersonal relationships, social context, and so forth. The generation of different-context cultures is related to factors such as economic methods, population density, history, and traditional culture. Healthcare providers can identify patients’ different communication needs based on the aforementioned factors and provide different communication modes. Current referral services do not mention the need for different communication styles based on contextual culture. Therefore, a study on contextual culture might become a future research direction of lung transplantation referral services, such as developing specific verbal tricks for different contexts and gradually forming communication norms through the application of verbal tricks.” (Page: 9; Line: 358)

“This study found two incidents that facilitated the patients’ referral decision-making process: one was submitting a referral application, and the other was receiving a lung transplantation suggestion from their doctor. Ivarsson et al. [28] used the critical incident technique (CIT) in a qualitative study to identify the information support, social support, and psychological support needs of patients in the waiting stage. In fact, CIT has been widely used to identify patients' decision-making needs in health services. Runeson et al. [29] used this technique to explore the factors influencing children's involvement in the decision-making process. Barradell et al. [30] used this technique to understand the pulmonary rehabilitation decision-making needs of patients with chronic obstructive pulmonary disease. Holden et al. [31] integrated critical incident and fictitious scenario techniques to create three distinct patient self-care decision-making modes to enhance current patient decision assistance programs. However, there are no studies that have used CIT to analyze these incidents that facilitate patients' lung transplantation referral decisions. Adding CIT to healthcare staff training or a "critical incident technician" to the team could assist healthcare staff in identifying and analyzing these incidents in patient referral decision-making process. However, the effectiveness of such measures needs to be further verified in quantitative studies. The development of a critical event checklist and package for a lung transplantation referral patient may become a hot topic for future research.” (Page: 10; Line: 379)

“Patients have different attitudes and use various information from peers, doctors, and so on in their decision-making process. Many studies obtained similar results but did not explore the causes and proposed more effective coping strategies [28,32]. The present study attempts to explain this phenomenon in the context of decision behavior, which may be explained by the different decision-making styles of patients. Schwartz et al. [33] indicated two different decision-making styles. Satisfying-style people tend to choose satisfactory decisions without making trade-offs, while maximizing-style people tend to compare each option to pursue the best decision [34]. In recent years, the classification of decision-making styles has usually used quantitative statistical methods [35], qualitative studies [36], or mixture studies [37] and has been used as a basis for developing user personas. The advantage of user personas is that hard-to-find user needs can be identified and thus provide more accurate e-health services [37]. With the global prevalence of coronavirus disease 2019, online referral services are a blessing for vulnerable patients with ALD. This suggests to us that user persona de-velopment will be a future research trend.” (Page: 10; Line: 397)

“This study identified several strategies that can improve decision-making self-efficacy during the patient's referral decision-making process: awakening confidence from the success of the decision-making processes of patients previously, providing transplantation referral decision experiences from peers, providing family attachment and oral approval responses, improving the ability to cope with lung transplantation uncertainty, and evoking the patient's expectations for future life. An investigation of patients with colorectal cancer [38] showed that higher decision self-efficacy was associated with lower decisional conflict. It can be seen that self-efficacy interventions can be added to early referral services to decrease decisional conflict, decisional regret, and late referral. Smith et al. [9] proposed measures in terms of making patients aware of the benefits of transplantation and misconceptions about the risks associated with transplantation in order to enhance patient self-efficacy. This supports some of the aforementioned strategies. In addition, the strategies proposed above are presented from additional perspectives, such as peers and family members, expanding on existing strategies to enhance patient self-efficacy in referral decision-making services. In the future, the aforementioned strategies can be transferred into a structured curriculum for patient health education or incorporated into referral decision-making services.”(Page: 10; Line: 411)

“This study is the first to explore the referral decision-making process for patients with ADL in China. Guided by a rigorous methodology, this study proposes several corresponding measures around four new findings. These measures will help improve the existing lung transplantation referral service component.” (Page: 11; Line: 429)

I also have the following more minor comments:

Point 2: For better clarity, I recommend numbering the four topics in the abstract.

Response 2: Thank you very much for your comment. We numbered the four topics in the abstract. This improves the readability of the abstract. (Page 1; Line 14)

Point 3: Why is it relevant that the interviewer was female? This information can be deleted or must be explained.

Response 3: Thank you very much for your comment. This information was deleted. (Page: 3; Line 126)

Point 4: Were the transcriptions anonymous?

Response 4: Thank you very much for your comment. The transcriptions were anonymous. A description was added to the remanuscript, as follows.

“Within 24 h of the interviews, the anonymized recordings were repeatedly and carefully listened to and then transcribed in Chinese verbatim by an undergraduate student who was not involved in this study. The first author and participants verified the correctness and completeness of the transcriptions. Then, the transcriptions were anonymized and imported into NVivo 12 Plus.” (Page 3; Line 134)

Point 5: In Table 3, capitalization is not consistent.

Response 5: Thank you very much for your comment. We modified the problem of inconsistent capitalization in the table.

Point 6: All ethics information is missing - which ethics committee approved this? In what way did the patients consent? What level of anonymity did this study have? Was there any impact on the patients' subsequent process?

Response 6: Thank you very much for your comment. We added the level of anonymity of this study and description impact on the patients' subsequent process. All ethical information is as follows:

  • Qusetion 1: Which ethics committee approved this?

“This study was performed in accordance with the Declaration of Helsinki and approved by the Medical Ethics Committee of Jiangnan University (JNU20210618IRB06).”(Page 2; Line 89)

  • Qusetion 2: In what way did the patients consent?

“The consent was provided in writing by willingness to participate.”(Page 3; Line 101)

  • Qusetion 3: What level of anonymity did this study have?

“Participants were informed of the study's purpose, study procedures, risks and benefits, confidentiality, and compensation, and that participation was voluntary. We assured participants that we would hide their identities by changing their names, changing the names of any other people they mentioned, etc., to ensure anonymity.” (Page 3; Line 98)”

  • Qusetion 4: Was there any impact on the patients' subsequent process?

“This study had no impact on the participants' subsequent processes.”(Page 3; Line124)

Point 7: Patient quotes should be more clearly highlighted as such.

Response 7: Thank you very much for your comment. To enhance the readability of this manuscript, we changed the format of the verbatim quotes after referring to the format of the qualitative research literature already published in International Journal of Environmental Research and Public Health.

Point 8: In the discussion, the authors should point more to comparable studies and place it in previous findings. In this context, it should be emphasized to what extent the study has novelty character.

Response 8: Thank you very much for your comment. We added highlighting the relevance of the results to the existing literature in the first paragraph of the discussion section and in each of the discussion points. In addition, we also clarified the innovative points of this study in "Strengths and Limitations". The following is the modified content (The highlighted places are the descriptions that were modified and related to the new insights.):

“This study explored the reasons behind patients' decisions to undertake lung transplantation referrals. The patient expectations of lung transplantation, access to information, and policy and social support were the reasons for patients' referral deci-sions and the need for early referral services, which is consistent with previous find-ings [8-10]. Additionally, there are four new findings in this study: (1) family attach-ment and oral approval response, (2) incidents leading to “It's decided then”, (3) dif-ferent patient attitudes and use of information, and (4) some strategies that can im-prove decision-making self-efficacy in lung transplantation referral decision-making process. The above findings could improve the existing lung transplantation referral service component.”(Page: 9; Line: 335)

“The findings of this study showed that some patients' decision making was based on family attachment and oral approval response. In the studies by Macdonald et al. [21] and Yelle et al. [22], only the importance of family companionship in patients was mentioned: there was no discussion of the two new forms of family support mentioned above.” (Page: 9; Line: 344)

“The roundabout expression reported in this study was not only an expression of a high-context culture but also a communication technique for patients to achieve their communication goals [26]. Hammami et al. [27] highlighted that the patients from different contextual cultures had different needs for information disclosure. When healthcare providers provide information to patients using communication that is not culturally appropriate to the patient's context, it can affect the patient's perception of the illness and the clinical assessment of the patient's psychosocial impairment, leading to late referrals [23]. In low-context cultures, communication must be clear and detailed to avoid distortion. In high-context cultures, communication is more focused on interpersonal relationships, social context, and so forth. The generation of different-context cultures is related to factors such as economic methods, population density, history, and traditional culture. Healthcare providers can identify patients’ different communication needs based on the aforementioned factors and provide different communication modes. Current referral services do not mention the need for different communication styles based on contextual culture. Therefore, a study on contextual culture might become a future research direction of lung transplantation referral services, such as developing specific verbal tricks for different contexts and gradually forming communication norms through the application of verbal tricks.” (Page: 9; Line: 358)

“This study found two incidents that facilitated the patients’ referral decision-making process: one was submitting a referral application, and the other was receiving a lung transplantation suggestion from their doctor. Ivarsson et al. [28] used the critical incident technique (CIT) in a qualitative study to identify the information support, social support, and psychological support needs of patients in the waiting stage. In fact, CIT has been widely used to identify patients' decision-making needs in health services. Runeson et al. [29] used this technique to explore the factors influencing children's involvement in the decision-making process. Barradell et al. [30] used this technique to understand the pulmonary rehabilitation decision-making needs of patients with chronic obstructive pulmonary disease. Holden et al. [31] integrated critical incident and fictitious scenario techniques to create three distinct patient self-care decision-making modes to enhance current patient decision assistance programs. However, there are no studies that have used CIT to analyze these incidents that facilitate patients' lung transplantation referral decisions. Adding CIT to healthcare staff training or a "critical incident technician" to the team could assist healthcare staff in identifying and analyzing these incidents in patient referral decision-making process. However, the effectiveness of such measures needs to be further verified in quantitative studies. The development of a critical event checklist and package for a lung transplantation referral patient may become a hot topic for future research.” (Page: 10; Line: 379)

“Patients have different attitudes and use various information from peers, doctors, and so on in their decision-making process. Many studies obtained similar results but did not explore the causes and proposed more effective coping strategies [28,32]. The present study attempts to explain this phenomenon in the context of decision behavior, which may be explained by the different decision-making styles of patients. Schwartz et al. [33] indicated two different decision-making styles. Satisfying-style people tend to choose satisfactory decisions without making trade-offs, while maximizing-style people tend to compare each option to pursue the best decision [34]. In recent years, the classification of decision-making styles has usually used quantitative statistical methods [35], qualitative studies [36], or mixture studies [37] and has been used as a basis for developing user personas. The advantage of user personas is that hard-to-find user needs can be identified and thus provide more accurate e-health services [37]. With the global prevalence of coronavirus disease 2019, online referral services are a blessing for vulnerable patients with ALD. This suggests to us that user persona de-velopment will be a future research trend.” (Page: 10; Line: 397)

“This study identified several strategies that can improve decision-making self-efficacy during the patient's referral decision-making process: awakening confidence from the success of the decision-making processes of patients previously, providing transplantation referral decision experiences from peers, providing family attachment and oral approval responses, improving the ability to cope with lung transplantation uncertainty, and evoking the patient's expectations for future life. An investigation of patients with colorectal cancer [38] showed that higher decision self-efficacy was associated with lower decisional conflict. It can be seen that self-efficacy interventions can be added to early referral services to decrease decisional conflict, decisional regret, and late referral. Smith et al. [9] proposed measures in terms of making patients aware of the benefits of transplantation and misconceptions about the risks associated with transplantation in order to enhance patient self-efficacy. This supports some of the aforementioned strategies. In addition, the strategies proposed above are presented from additional perspectives, such as peers and family members, expanding on existing strategies to enhance patient self-efficacy in referral decision-making services. In the future, the aforementioned strategies can be transferred into a structured curriculum for patient health education or incorporated into referral decision-making services.”(Page: 10; Line: 411)

“This study is the first to explore the referral decision-making process for patients with ADL in China. Guided by a rigorous methodology, this study proposes several corresponding measures around four new findings. These measures will help improve the existing lung transplantation referral service component.” (Page: 11; Line: 429)

Reviewer 2 Report

From the title of the article, I expected that it would be a study
of natural data for the purpose of linguistic interpretation.
The authors submit a publication in which they somehow
present the observed data related to lung transplantation research.

The publication does not mention any method by which the authors
interpret the data using natural language sentences.

For example, the authors claim that

Almost all participants regarded lung transplantation as a gamble.
Most participants described that they were eager to achieve “normal life.”

How did the authors figure out that ``Almost all'' participants.....

We can come up with a whole range of such statements.
Can the authors somehow explain where they came up with these claims.
What do the abbreviations in parentheses mean? (P24)????

I am an expert in data interpretation using natural language
expressions using intermediate quantifiers.
There are several mathematical models to mathematically define
these expressions. The authors should therefore support the
study with some mathematical model.

Author Response

Response to Reviewer 2 Comments

Point 1: From the title of the article, I expected that it would be a study of natural data for the purpose of linguistic interpretation. The authors submit a publication in which they somehow present the observed data related to lung transplantation research.

Response 1:

First of all, we thank the reviewer for the suggestions and comments. As you mentioned, we used a qualitative study to present retrospective experiences and perspectives of the lung transplantation referral stage for patients with advanced lung disease (ALD). To explore the reasons behind patients receiving lung transplantation referrals.

According to our understanding of linguistic interpretation, I am afraid that qualitative research has different research objectives and methods compared to the linguistic interpretation you mentioned. We are not sure if the language interpretation you mention is in the field of computer science? Computer technology and modeling techniques are usually used in linguistic interpretation [1]. However, qualitative research is a research method in the field of sociology. Qualitative research aims to provide in-depth insights and understanding of real-world problems and, in contrast to quantitative research, it does not introduce treatments, manipulate or quantify predefined variables [2]. Qualitative research at its core, ask open-ended questions whose answers are not easily put into numbers such as ‘how’ and ‘why’[2]. Apparently, qualitative research is the more appropriate research method according to the purpose of this study(Purpose of this study: To explore the reasons for patients receiving lung transplantation referrals.).

Point 2: The publication does not mention any method by which the authors interpret the data using natural language sentences. For example, the authors claim that Almost all participants regarded lung transplantation as a gamble. Most participants described that they were eager to achieve “normal life.” How did the authors figure out that ``Almost all'' participants.....We can come up with a whole range of such statements. Can the authors somehow explain where they came up with these claims.

Response 2:

Thank you very much for your comment. Content analysis is a method that may be used with either qualitative or quantitative data [3]. The content analysis method used in this qualitative study is the conventional content analysis method widely accepted in sociological and nursing research [3]. It is not appropriate to present numbers/percentages in the results derived from conventional content analysis methods. We modified some indication would help the reader appreciate which were widely held opinions and which were minority perspectives.

We listed a more complete data analysis process below, and we believe you will understand why a number/percentage is inappropriate after you understand the process of conventional content analysis.

As mentioned in the Materials and Methods section (Page 2; Line: 78), this study used conventional content analysis to analyze and interpret the reported verbal data [4]. Two authors repeated reading and immersion in the material. These two authors separately open-coded, created categories, and conceptualized the material after ensuring com-plete familiarity with the text and understanding of the data. To clarify the process of this analysis, we will describe it in 3 steps:

  • Open-coded: Two authors began by reading each transcript from beginning to end, as one would read a novel. Then, they separately read each transcription carefully, highlighting text that appeared to describe lung transplantation referral decision-making process and writing a keyword or phrase that seemed to capture the emotional reaction, using the participant's words. After open coding of three to four transcripts, two authors decided on preliminary codes. The disagreements were discussed with a third researcher until all three researchers agreed. They then coded the remaining transcriptions (and recoded the original ones) using these codes and adding new codes when they encountered data that did not fit into an existing code. Disagreements arising from this process still need to be discussed with a third author until all three researchers agreed. Once all transcripts had been coded, two authors examined all data within a particular code.

  • Created categories, and conceptualized the material: In the above process, some codes are combined, while others are divided into sub-themes. As mentioned in the manuscript, a preliminary theme, subthemes, and their concepts were developed after the three researchers reached a consensus.

  • Finally, the themes, subthemes, and concepts were reviewed with some of the participants and the research team of eight researchers to reduce biased interpretations and ensure trustworthiness. The research team included both males and females, and had rich experience in qualitative research in respiratory care.

Point 3: What do the abbreviations in parentheses mean? (P24)????

Response 3: Thank you very much for your comment. We added the meaning of this abbreviation to the manuscript as follows:

“Participants were numbered according to interview order (Patients: P1, P2…)”(Page 4;Line 138)

Point 4: I am an expert in data interpretation using natural language expressions using intermediate quantifiers. There are several mathematical models to mathematically define these expressions. The authors should therefore support the study with some mathematical model.

Response 4: Thank you very much for your advice from the perspective of data interpretation. And it is interesting to define expressions in terms of mathematical models. Your suggestion is indeed broadening our horizon. However, it is clearly inappropriate from the methodological point of view of this study.

References

  • Gelbukh A, Kolesnikova O. Linguistic Interpretation. In: Gelbukh A, Kolesnikova O, editors. Semantic Analysis of Verbal Collocations with Lexical Functions. Berlin, Heidelberg: Springer Berlin Heidelberg; 2013. p. 85-92. https://doi.org/10.1007/978-3-642-28771-8_6
  • Moser A, Korstjens I. Series: Practical guidance to qualitative research. Part 1: Introduction. Eur J Gen Pract. 2017;23(1):271-273. http://doi:10.1080/13814788.2017.1375093
  • Elo S, Kyngäs H. The qualitative content analysis process. J Adv Nurs. 2008;62(1):107-115. https://doi.org/10.1111/j.1365-2648.2007.04569.x
  • Hsieh HF, Shannon SE. Three approaches to qualitative content analysis. Qual Health Res. 2005;15(9):1277-1288. https://doi.org/10.1177/1049732305276687

Reviewer 3 Report

This is an interesting and useful paper which reports from a fairly substantial qualitative study. The methods section is thorough and covers the essential information required.

 However, overall I think the number of limitations to the paper makes it difficult for me to recommend publication  as there is a lot of work that would need to be done to bring the manuscript up to the required level. These are my main comments:

 The aims of the study/paper could be more clearly articulated, i.e. here

‘This study explored the reasons for the lung transplantation referral decisions of patients based on retrospective experience and perspectives of the referral stage of patients with advanced lung disease (ALD) in China. It also aimed to provide strategies for the early referral of potential lung transplant candidates with high-quality information services and comprehensive decision support so as to reduce delayed referrals and so on’

 At times the written English could be improved in terms of word selection and grammar. In general this isn’t a major problem but the meaning is not clear in places (e.g. ‘the ambiguity was further confirmed during the interview…’ p. 3)

 The organisation and presentation of findings needs work in my view. The numbering and headers detract from the flow (e.g. headers 3.3. and 3.3.1 are nearly identical), and it is hard to distinguish verbatim quotes from the text written by the authors. Some of the headers are unclear in meaning (e.g. 3.2.1. bottom of page 5).

 I can only speak as a UK-based scholar but I think the term ‘critical incident’ is likely to be misleading.  I don’t think the literature deriving from health and criminal justice would include a conversation with your doctor as a critical incident.

 The findings section doesn’t give an indication of the prevalence of each theme in the dataset.  Clearly it is not appropriate to present figures/percentages but some indication would help the reader appreciate which were widely held opinions and which were minority perspectives.

 The discussion is too definitive in its recommendation of e.g. ‘critical incident techniques’.  It’s fine to float some possible interventions but the study itself doesn’t warrant the strength with which these are put forward. I think the study represents a first stage towards eventually recommending some evidence-based interventions, but it is too early to make these kinds of definitive recommendations.

The literature introduced in the discussion section is interesting but a little arbitrary and runs the risk of incoherence (e.g. from Habermas to cluster analysis!)

Minor points:

 The authors refer to ‘consolidated reports of qualitative research’. Do they mean Consolidated criteria for reporting qualitative research (COREQ)? If so, the proper source should be cited

The authors refer to a ‘literature study’ (p.3) but it is not clear what this is

Authors use terms ‘sex’ and later ‘gender’ – it isn’t clear if these are intended to refer to different things

Can the authors clarify how the recordings were ‘transferred into text’? (p.3)

Author Response

Response to Reviewer 3 Comments

This is an interesting and useful paper which reports from a fairly substantial qualitative study. The methods section is thorough and covers the essential information required.

However, overall I think the number of limitations to the paper makes it difficult for me to recommend publication as there is a lot of work that would need to be done to bring the manuscript up to the required level.

Response:First of all, thank you very much for your recognition of the methodological aspects of this study. We are also really appreciate your numerous suggestions for improvement! According to your comments, we made major revisions to the manuscript's study objectives, results, and discussion sections. Additionally, to improve the written English of the manuscript, we have used the English editing services of MDPI. We hope that our revisions to the manuscript bring the manuscript up to the required level. Below are our responses to each comment and the details of our modifications. We marked page and line number in parentheses.

These are my main comments:

Point 1: The aims of the study/paper could be more clearly articulated, i.e. here

‘This study explored the reasons for the lung transplantation referral decisions of patients based on retrospective experience and perspectives of the referral stage of patients with advanced lung disease (ALD) in China. It also aimed to provide strategies for the early referral of potential lung transplant candidates with high-quality information services and comprehensive decision support so as to reduce delayed referrals and so on’

Response 1: Thank you very much for your comment. We improved the statement of the aims of the study to make it more clearly articulated. The following is the modified content:

“This study was based on retrospective experiences and perspectives of the lung transplantation referral stage for patients with advanced lung disease (ALD) in China to understand the decision-making process of patients when accepting referrals. This study aimed to explore the reasons behind patients receiving lung transplantation re-ferrals and provide evidence to support the improvement of existing referral service components.”(Page 2; Line 73)

Point 2: At times the written English could be improved in terms of word selection and grammar. In general this isn’t a major problem but the meaning is not clear in places (e.g. ‘the ambiguity was further confirmed during the interview…’ p. 3)

Response 2: Thank you very much for your comment. To improve the written English of the manuscript, we have used the English editing services of MDPI. Additionally, in response to your question about the meaning here (e.g. 'the ambiguity was further confirmed during the interview...' p. 3), we have improved the expression. The following is the modified content:

“During the interview, when the participant's statements were unclear, the interviewer would further confirm the statement.” (Page 3; Line 112)

Point 3: The organisation and presentation of findings needs work in my view. The numbering and headers detract from the flow (e.g. headers 3.3. and 3.3.1 are nearly identical), and it is hard to distinguish verbatim quotes from the text written by the authors. Some of the headers are unclear in meaning (e.g. 3.2.1. bottom of page 5).

Response 3: Thank you very much for your comment. We made modifications to the organisation and presentation of the study results. Here below you can find our responses to each question in this part.

  • Qusetion 1: The numbering and headers detract from the flow (e.g. headers 3.3. and 3.3.1 are nearly identical).

Thank you very much for your comment. In order to clarify and make headers 3.3. and 3.3.1 easier for the reader to understand, we have revised the theme statement of 3.3. Instead of 'Different attitudes toward, and of, information', we changed the 3.3 to 'Facing various information from peers, doctors, and so on'. And we improved the description of 3.3. None of these changes have altered the original meaning, but have simply made the expression easier to understand. The following is the modified content:

“3.3 Facing various information from peers, doctors, and so on

All the participants reported their attitudes toward, and use of, information when they faced various information from peers, doctors, and so on during the decision-making process.”(Page 7;Line 240)

  • Qusetion 2: It is hard to distinguish verbatim quotes from the text written by the authors.

Thank you very much for your comment. To enhance the readability of this manuscript, we changed the format of the verbatim quotes after referring to the format of the qualitative research literature already published in International Journal of Environmental Research and Public Health.

  • Qusetion 3: Some of the headers are unclear in meaning (e.g. 2.1. bottom of page 5).

Thank you very much for your comment. In order to clarify the meaning of the header of 3.2.1., we changed the header of 3.2.1 from "Possibility of decision making change being fear" to " Hesitation when making the decision due to fear". And we improved the description. None of these changes have altered the original meaning, but have simply made the expression easier to understand. The following is the modified content:

“3.2.4. Hesitation when making the decision due to fear

A minority of participants described their hesitation due to fear. Participants in-dicated that they might finally refuse the lung transplantation, even if they decided to accept the referral at first and had been placed on the waiting list.” (Page 6;Line 232)

Additionally, in order to maintain the flow of 3.2 (Facing uncertain outcomes), we changed the order of the sub-themes to make them look more flowing. The following are the details of our modifications:

3.2. Facing uncertain outcomes

3.2.1 Personal luck arranging everything

3.2.2 Belief in success

3.2.3 Incidents leading to “It’s decided then”

3.2.4 Hesitate to make the decision being fear

In addition to the places you specifically mentioned above, we have reviewed all the results in terms of the fluency of the headings and the clarity of the presentation. We improved those descriptions of the themes. None of these changes have altered the original meaning, but have simply made the expression easier to understand. Additionally, For easier reading and understanding, we have revised the numbers of the themes in the table to match the numbers in the text. The following is the modified content:

“3.2 Facing uncertain outcomes

The majority of participants expressed a fear of uncertain outcomes. Some participants did not fear uncertain outcomes and expressed their determination in decision making. However, some participants expressed a fear of uncertain outcomes and reported hes-itation in their decision-making process.”(Page 6;Line 196)

“3.2.3. Incidents leading to “It's decided then”

The majority of participants reported that they could make the decision calmly after some incidents. Participants reported two types of incidents.” (Page 6;Line 219)

“3.4 Complex policy and societal support

All the participants reported that policy and societal support had a significant impact on their referral decisions.” (Page 8;Line 285)

The following is the modified table:

Themes

Subthemes

3.1. Expectations for lung transplantation leading to the decision

3.1.1 A gamble for a silver lining

3.1.2 Back to normal life

3.1.3 Back to occupation

3.2. Facing uncertain outcomes

3.2.1 Personal luck arranging everything

3.2.2 Belief in success

3.2.3 Incidents leading to “It’s decided then”

3.2.4 Hesitate to make the decision being fear

3.3. Facing various information from peers, doctors, and so on

3.3.1 Different attitudes toward information

3.3.2 Use of information to weigh risks

3.4. Complex policy and society support

3.4.1 Provided earlier transplantation referral services

3.4.2 Family attachment and oral approval response contributed to the referral decision

3.4.3 Getting financial support from medical insurance and multi-welfare

Point 4: I can only speak as a UK-based scholar but I think the term ‘critical incident’ is likely to be misleading. I don’t think the literature deriving from health and criminal justice would include a conversation with your doctor as a critical incident.

Response 4: Thank you very much for your comment. We agree that your reference to the term " critical incidents" may be misleading.Therefore, we changed the theme to "Incidents leading to ‘It’s decided then’". And we improved the description to make it easier for the reader to understand. We hope that the current description will clarify the misunderstandings. The following is the modified content:

“3.2.3. Incidents leading to ‘It's decided then’

The majority of participants reported that they could make the decision calmly after some incidents. Participants reported two types of incidents.”(Page 6; Line 219)

Point 5: The findings section doesn’t give an indication of the prevalence of each theme in the dataset.  Clearly it is not appropriate to present figures/percentages but some indication would help the reader appreciate which were widely held opinions and which were minority perspectives.

Response 5: Thank you very much for your comment. We have changed the words 'most', 'some' and so on from the original expressions to a uniform 'majority' and 'minority'. We hope that this would help the reader appreciate which were widely held opinions and which were minority perspectives.

Point 6: The discussion is too definitive in its recommendation of e.g. ‘critical incident techniques’.  It’s fine to float some possible interventions but the study itself doesn’t warrant the strength with which these are put forward. I think the study represents a first stage towards eventually recommending some evidence-based interventions, but it is too early to make these kinds of definitive recommendations.

Response 6: Thank you very much for your comment. Your comments have largely enhanced the rigour of the discussion. We added a description of the current status of the use of CIT in lung transplant patients. It is also noted that the measures proposed in this study need to be further validated for feasibility in quantitative studies. Additionally, we also added this point into the section of “Strengths and Limitations”. The following is the modified content (The highlighted places are the descriptions that were modified according to your comments):

“This study found two incidents that facilitated the patients’ referral deci-sion-making process: one was submitting a referral application, and the other was receiving a lung transplantation suggestion from their doctor. Ivarsson et al. [28] used the critical incident technique (CIT) in a qualitative study to identify the information support, social support, and psychological support needs of patients in the waiting stage. In fact, CIT has been widely used to identify patients' decision-making needs in health services. Runeson et al. [29] used this technique to explore the factors influenc-ing children's involvement in the decision-making process. Barradell et al. [30] used this technique to understand the pulmonary rehabilitation decision-making needs of patients with chronic obstructive pulmonary disease. Holden et al. [31] integrated critical incident and fictitious scenario techniques to create three distinct patient self-care decision-making modes to enhance current patient decision assistance programs. However, there are no studies that have used CIT to analyze these incidents that facilitate patients' lung transplantation referral decisions. Adding CIT to healthcare staff training or a "critical incident technician" to the team could assist healthcare staff in identifying and analyzing these incidents in patient referral decision-making process. However, the effectiveness of such measures needs to be further verified in quantitative studies. The development of a critical event checklist and package for a lung transplantation referral patient may become a hot topic for future research.” (Page 10; Line 378)

“As an exploratory study, the measures proposed in this study still need experimental studies to further verify their effectiveness.” (Page 11; Line 432)

Point 7: The literature introduced in the discussion section is interesting but a little arbitrary and runs the risk of incoherence (e.g. from Habermas to cluster analysis!)

Response 7: We are very grateful for your positive comments on our discussion section. In addition, according to your comment, we made modifications to improve the coherence of the discussion section. The following are the details of the improvements:

“The roundabout expression reported in this study was not only an expression of a high-context culture but also a communication technique for patients to achieve their communication goals [26]. Hammami et al. [27] highlighted that the patients from different contextual cultures had different needs for information disclosure. When healthcare providers provide information to patients using communication that is not culturally appropriate to the patient's context, it can affect the patient's perception of the illness and the clinical assessment of the patient's psychosocial impairment, lead-ing to late referrals [23]. In low-context cultures, communication must be clear and detailed to avoid distortion. In high-context cultures, communication is more focused on interpersonal relationships, social context, and so forth. The generation of different-context cultures is related to factors such as economic methods, population density, history, and traditional culture. Healthcare providers can identify patients’ different communication needs based on the aforementioned factors and provide different communication modes. Current referral services do not mention the need for different communication styles based on contextual culture. Therefore, a study on contextual culture might become a future research direction of lung transplantation referral ser-vices, such as developing specific verbal tricks for different contexts and gradually forming communication norms through the application of verbal tricks. Moreover, the training of healthcare providers should be enhanced so that healthcare providers can correctly identify the potential goals and needs of the patients during the communica-tion process and provide personalized assistance to ensure the quality of service.”(Page 9; Line: 357)

“This study found two incidents that facilitated the patients’ referral deci-sion-making process: one was submitting a referral application, and the other was re-ceiving a lung transplantation suggestion from their doctor. Ivarsson et al. [28] used the critical incident technique (CIT) in a qualitative study to identify the information support, social support, and psychological support needs of patients in the waiting stage. In fact, CIT has been widely used to identify patients' decision-making needs in health services. Runeson et al. [29] used this technique to explore the factors influenc-ing children's involvement in the decision-making process. Barradell et al. [30] used this technique to understand the pulmonary rehabilitation decision-making needs of patients with chronic obstructive pulmonary disease. Holden et al. [31] integrated critical incident and fictitious scenario techniques to create three distinct patient self-care decision-making modes to enhance current patient decision assistance pro-grams. However, there are no studies that have used CIT to analyze these incidents that facilitate patients' lung transplantation referral decisions. Adding CIT to healthcare staff training or a "critical incident technician" to the team could assist healthcare staff in identifying and analyzing these incidents in patient referral deci-sion-making process. However, the effectiveness of such measures needs to be further verified in quantitative studies. The development of a critical event checklist and package for a lung transplantation referral patient may become a hot topic for future research.”

(Page: 10; Line: 378 )

“Patients have different attitudes and use various information from peers, doctors, and so on in their decision-making process. Many studies obtained similar results but did not explore the causes and proposed more effective coping strategies [28,32]. The present study attempts to explain this phenomenon in the context of decision behavior, which may be explained by the different decision-making styles of patients. Schwartz et al. [33] indicated two different decision-making styles. Satisfying-style people tend to choose satisfactory decisions without making trade-offs, while maximizing-style people tend to compare each option to pursue the best decision [34]. In recent years, the classification of decision-making styles has usually used quantitative statistical methods [35], qualitative studies [36], or mixture studies [37] and has been used as a basis for developing user personas. The advantage of user personas is that hard-to-find user needs can be identified and thus provide more accurate e-health services [37]. With the global prevalence of coronavirus disease 2019, online referral services are a blessing for vulnerable patients with ALD. This suggests to us that user persona de-velopment will be a future research trend.” (Page 10; Line: 396)

Minor points:

Point 8: The authors refer to ‘consolidated reports of qualitative research’. Do they mean Consolidated criteria for reporting qualitative research (COREQ)? If so, the proper source should be cited

Response 8: Thank you very much for your comment. We modified this point and added relevant references. The following are the details of the improvements:

“The study followed Consolidated criteria for reporting qualitative research (COREQ) to ensure quality and transparency [18].” (Page 2; Line: 86)

[18] Tong A, Sainsbury P, Craig J. Consolidated criteria for reporting qualitative research (COREQ): a 32-item checklist for inter-views and focus groups. Int J Qual Health Care. 2007;19(6):349-357. http://doi:10.1093/intqhc/mzm042 (Page 12; Line: 523)

Point 9: The authors refer to a ‘literature study’ (p.3) but it is not clear what this is.

Response 9: Thank you very much for your comment. We explain 'literary study' further. The following are the details of the improvements:

“Based on a review of the literature on qualitative studies of the experiences and perspectives of patients in the referral, evaluation, and listing stage, the research team worked together to construct a semi-structured interview guideline.” (Page 3; Line: 107)

Point 10: Authors use terms ‘sex’ and later ‘gender’ – it isn’t clear if these are intended to refer to different things

Response 10: Thank you very much for your comment. We changed the word 'sex' to 'gender' in one place in the manuscript. (Page 3; Line: 105)

Point 11: Can the authors clarify how the recordings were ‘transferred into text’? (p.3)

Response 11: Thank you very much for your comment. We further clarify the process of text transcription. The following is the modified content:

“Within 24 h of the interviews, the anonymized recordings were repeatedly and carefully listened to and then transcribed in Chinese verbatim by an undergraduate student who was not involved in this study. The first author and participants verified the correctness and completeness of the transcriptions. Then, the transcriptions were anonymized and imported into NVivo 12 Plus. Participants were numbered according to the interview order (Patients: P1, P2…)” (Page 3; Line: 134)

Round 2

Reviewer 1 Report

The authors have worked on the criticized aspects and the manuscript is now much improved.

Author Response

Thank you very much for your recognition of our revised manuscript.

Reviewer 3 Report

Thanks, the authors have addressed my queries.  Here are a few minor corrections I spotted on reading the second version:

Page 1: can the authors change ‘Medicaid insurance status’ to a more universal term such as ‘insurance status’? most countries don't have medicaid but do have some kind of insurance (either social or private)

Unless I missed it, there should probably be a quick statement of when/how the data were translated into English language (page 3-4)

p.6 i suggest rephrasing ‘The majority of participants described that “getting lung transplantation suggestion from their doctor” was an incident for them to make the decision calmly’ to “getting lung transplantation suggestion from their doctor” was an incident which enabled them to make the decision calmly’

p.8 suggest rephrasing  ‘Additionally, the majority of participants reported that the response of family members to their own decision could promote their decision-making process’ to ‘Additionally, the majority of participants reported that the response of family members to their own decision could influence their decision-making process’

p.9 I think ‘The roundabout expression’ needs to be explained
